# *Cydonia oblonga*-Seed-Mucilage-Based pH-Sensitive Graft Copolymer for Controlled Drug Delivery—In Vitro and In Vivo Evaluation

**DOI:** 10.3390/pharmaceutics15102445

**Published:** 2023-10-10

**Authors:** Muhammad Sarfraz, Ume Ruqia Tulain, Alia Erum, Nadia Shamshad Malik, Arshad Mahmood, Sidra Aslam, Mansur Abdullah Sandhu, Muhammad Tayyab

**Affiliations:** 1College of Pharmacy, Al Ain University, Al Ain Campus, Al Ain 64141, United Arab Emirates; 2Faculty of Pharmacy, University of Sargodha, Punjab 40100, Pakistan; alia.erum@uos.edu.pk (A.E.); sidraaslum@gmail.com (S.A.); 3Faculty of Pharmacy, Capital University of Science and Technology, Islamabad 44000, Pakistan; nadia.malik@cust.edu.pk; 4College of Pharmacy, Al Ain University, Abu Dhabi Campus, Abu Dhabi 64141, United Arab Emirates; arshad.mahmood@aau.ac.ae; 5AAU Health and Biomedical Research Center, Al Ain University, Abu Dhabi 64141, United Arab Emirates; 6Faculty of Pharmacy and Alternative Medicine, The Islamia University, Bahawalpur 63100, Pakistan; sumairak857@gmail.com; 7Department of Physiology, Phir Meher Ali Shah Arid Agriculture University, Rawalpindi 46000, Pakistan; mansoorsandhu@uaar.edu.pk; 8Department of Pharmacy, Quaid Azam University, Islamabad 15320, Pakistan; mtayyab@bs.qau.edu.pk

**Keywords:** pH responsive, *Cydonia oblonga* mucilage, hydrogel, graft copolymer, acrylic acid, methacrylic acid

## Abstract

The primary objective of this study was to assess the potential utility of quince seed mucilage as an excipient within a graft copolymer for the development of an oral-controlled drug delivery system. The *Cydonia oblonga*-mucilage-based graft copolymer was synthesized via a free radical polymerization method, employing potassium per sulfate (KPS) as the initiator and N, N-methylene bisacrylamide (MBA) as the crosslinker. Various concentrations of monomers, namely acrylic acid (AA) and methacrylic acid (MAA), were used in the graft copolymerization process. Metoprolol tartarate was then incorporated into this graft copolymer matrix, and the resultant drug delivery system was subjected to comprehensive characterization using techniques such as Fourier-transform infrared spectroscopy (FTIR) and scanning electron microscopy (SEM). The swelling behavior of the drug delivery system was evaluated under different pH conditions, and in vitro drug release studies were conducted. Furthermore, pharmacokinetic parameters including the area under the curve (AUC), maximum plasma concentration (Cmax), time to reach Cmax (Tmax), and half-life (t1/2) were determined for metoprolol-loaded hydrogel formulations in rabbit plasma, and these results were compared with those obtained from a commercially available product. The key findings from the study include observations that higher concentrations of acrylic acid (AA) and *Cydonia oblonga* mucilage (CM) in the graft copolymer enhanced swelling, while the opposite trend was noted at elevated concentrations of methacrylic acid (MAA) and N, N-methylene bisacrylamide (MBA). FTIR analysis confirmed the formation of the graft copolymer and established the compatibility between the drug and the polymer. SEM imaging revealed a porous structure in the prepared formulations. Additionally, the swelling behavior and drug release profiles indicated a pH-sensitive pattern. The pharmacokinetic assessment revealed sustained release patterns of metoprolol from the hydrogel network system. Notably, the drug-loaded formulation exhibited a higher Cmax (156.48 ng/mL) compared to the marketed metoprolol product (96 ng/mL), and the AUC of the hydrogel-loaded metoprolol was 2.3 times greater than that of the marketed formulation. In conclusion, this study underscores the potential of quince seed mucilage as an intelligent material for graft-copolymer-based oral-controlled release drug delivery systems.

## 1. Introduction

Pharmaceutical scientists have long been dedicated to developing precise drug delivery systems that can effectively transport bioactive agents within the body [1]. Among the various materials considered, hydrogels, with their versatile polymeric 3D framework, have gained prominence in the field of controlled drug delivery. Hydrogels can be categorized as synthetic or natural, and their exceptional properties make them promising candidates for overcoming the limitations of conventional oral drug delivery, which includes challenges such as drug degradation in the gastrointestinal tract and short drug half-lives [2]. Recent advancements in hydrogel synthesis have resulted in the development of smart hydrogels capable of responding to external stimuli like changes in pH, temperature, light exposure, and ionic strength, making them highly attractive for drug delivery applications [3].

Hydrogels, with their intricately crosslinked structures and high water content, closely mimic natural soft tissues, making them particularly well suited for drug delivery [4]. By carefully manipulating polymeric networks, hydrogels can be tailored to interact with their environment in a controlled manner, with pH-responsive hydrogel systems being especially significant [5]. These systems can release their cargo in response to fluctuations in pH, aligning perfectly with the dynamic requirements of the biological milieu [6]. Polymeric drug carriers have the intention of developing their therapeutic efficacy and decreasing toxicity. Natural polymers, especially polysaccharides, have gained attention in drug delivery due to their favorable biological properties, low toxicity, and biocompatibility [7,8]. Their availability, eco-friendliness, structural diversity, and in vivo biodegradability further enhance their suitability as matrix materials in “smart” drug delivery systems [9].

The pH-responsive behavior of natural polysaccharides is a key factor in their effectiveness in drug delivery systems. This behavior arises from the presence of ionizable functional groups, such as hydroxyl and carboxyl groups, within the polysaccharide structure [10]. In an acidic environment, such as the stomach, these polysaccharides undergo protonation, leading to increased chain flexibility and swelling of the polymer matrix. Conversely, in a more alkaline environment, like the small intestine, deprotonation occurs, resulting in reduced chain flexibility and deswelling of the polymer. This inherent pH sensitivity allows for precise control over drug release kinetics, as drugs encapsulated within the polysaccharide matrix are released selectively in response to the pH conditions encountered along the gastrointestinal tract [11]. This pH-dependent behavior, coupled with the biocompatibility and biodegradability of natural polysaccharides, positions them as scientifically robust candidates for developing “smart” drug delivery systems with enhanced precision [12].

*Cydonia oblonga* mucilage, a naturally occurring hydrogel, has generated interest in advanced drug delivery systems due to its advantages over synthetic polymers [13]. This naturally occurring swellable polysaccharide offers distinct advantages, particularly its exceptional swelling properties, including pH- and saline-responsive swelling–deswelling capabilities [14]. Cydonia oblonga seed mucilage is mostly composed of glucuronic acid and xylose, which makes it a biocompatible polysaccharide. Cydonia oblonga seed mucilage is water-swellable, so it potentially acts as a smart material. It usually deswells in acidic conditions and swells in water or basic buffers. These novel characteristics are augmented by crosslinking with acrylic acid and methacrylic acid [15].

Metoprolol tartrate (MT), a representative of BCS class 1 drugs known for their antihypertensive properties, serves as a model drug for our study. MT’s predictable pharmacokinetic behavior and pharmacological profile align with the goals of controlled drug release. Acting as a cardio-selective competitive beta-1 adrenergic receptor antagonist, MT plays a pivotal role in managing cardiovascular conditions. However, its relatively short half-life and limited bioavailability due to first-pass metabolism necessitate the development of controlled-release formulations to maintain therapeutic levels in the bloodstream and enhance efficacy [15].

In the realm of hydrogels, the selection of monomers holds paramount importance, as it profoundly influences fundamental aspects such as swelling characteristics, drug encapsulation capacity, and drug release kinetics [16]. The interplay between polymers and monomers within hydrogels encompasses various interactions, including physical entanglement, hydrogen bonding, van der Waals forces, and covalent bonding. Monomers such as acrylic acid and methacrylic acid, known for their pH-responsive attributes, are particularly noteworthy, with the carboxylic acid moiety significantly contributing to ionic repulsion dynamics [17].

The central objective of this scientific inquiry revolves around the extraction of mucilage from Cydonia oblonga seeds and the subsequent formulation of a mucilage-based hydrogel. This comprehensive investigative endeavor encompasses rigorous in vitro and in vivo assessments of the mucilage-based hydrogel, marking a significant advancement in the field of controlled drug delivery systems.

## 2. Materials and Methods

### 2.1. Materials

*Cydonia oblonga* mucilage was obtained from the local market in Sargodha. N,N-methylene bisacrylamide, acrylic acid, sodium hydroxide, hydrochloric acid, and potassium dihydrogen phosphate were obtained from Merck Germany. Potassium persulfate was purchased from Sigma Aldrich, St. Louis, MO, USA. Additionally, acetonitrile, methanol, and ammonium acetate buffer were obtained from Honeywell Riedel-de-Haen, Germany.

### 2.2. Extraction of Cydonia oblonga Mucilage

Dried *Cydonia oblonga* mucilage was obtained by removing the broken *Cydonia* seeds via screening, and 200 g of seeds were soaked in 800 mL of water for 8 h before heating at 60 °C for 1–2 h. Extruded mucilage was separated by using a cotton cloth and washed thrice with n-hexane to remove lipophilic substances. Afterwards, it was dried in hot air in an oven at 45 °C. Dried mucilage was ground with a pestle and mortar and sieved by using an 80-mesh sieve [18].

### 2.3. Preparation of Cydonia oblonga-Mucilage-Based Hydrogels

*Cydonia mucilage* (CM) hydrogels were prepared by the free radical polymerization method. Different formulations were prepared by using varying ratios of three different monomers, the crosslinker, and different ratios of polymer, as shown in Table 1 and Table 2. For formulation of the hydrogel polymer, a solution was prepared by dissolving *Cydonia mucilage* (CM) in distilled water at 70 °C on a hot plate with continuous stirring with a magnetic stirrer. As the initiator, potassium per sulphate was slowly added in a polymer solution under continuous stirring; the mixture was cooled to room temperature. Then, the crosslinker was dissolved in the monomer with the help of a magnetic stirrer. Then, this solution was poured into the polymer solution with continuous stirring at ambient temperature. The final weight of this hydrogel preparation was made up to 100 g with distilled water and stirred for 2 min; then, it was placed in a water bath at 55 °C for 1 h, then 65 °C for another hour, and, finally, kept at 80 °C until completion of the reaction. Bubble formation was avoided by a gradual increase in the temperature. After completion of the reaction, hydrogels were removed from the water bath and cooled to room temperature. The formulated hydrogel was cut into disks of 0.5 cm thickness with the help of a sharp cutter. These disks were washed with an excess of 30% ethanol: water solution in distilled water for 24 h to remove un-reacted monomer and excess of reagents. After 24 h, the ethanol solution was changed to a fresh solution for another 24 h. After complete washing, hydrogel disks were oven dried at 50 °C until a constant weight of the hydrogels was obtained. The dried disks were stored in vacuum desiccator for characterization [19].

### 2.4. Physicochemical Properties of Hydrogels

#### 2.4.1. Swelling Studies

The dried disks were weighed and immersed in USP buffer solutions of pH 1.2 and pH 7.4. The samples were taken out of the buffer solutions at regular time intervals, and excess surface water was removed by blotting using laboratory tissue before weighing. After achieving dynamic swelling, the samples remained in the same solutions and were used for equilibrium swelling [20].

The dynamic and equilibrium swelling ratios of different formulations were found by using following equations:(1)$$q=W_{h}/W_{d}$$
where *W_d_* shows the weight of the dried hydrogel disk; “*q*” represents dynamic swelling; and *W_h_* indicates the swollen gel’s weight at time *t* [21].

Equivalent weight was used for the determination of percent equilibrium swelling (%*ES*) of each formulation by using the following equation:(2)$$\%\;ES=\frac{Meq-M_{0}}{Meq}\times 100$$
where *M*_0_ and *M_eq_* indicate the mass of dried and swollen gel disks at equilibrium, respectively. The swelling ratio and percent equilibrium swelling were determined according to the equations [21].

#### 2.4.2. Drug Loading

The swelling diffusion method for loading hydrogel disks was used. A 1% drug solution in phosphate buffer of pH 7.4 was made. One disk of every formulation was immersed in 100 mL of 1% of the drug solution until swelling equilibrium was reached. Disks were taken out of the solution after achieving swelling equilibrium; excess surface drug was removed by washing with distilled water. First, it was allowed to dry at room temperature and then oven dried at 50 °C until the drying equilibrium was achieved [20].

The amount of loaded drugs in the disks were measured by following formula:Total drug loaded = *W_L_ − W_U_*(3)
where *W_L_* and *W_U_* are weight of dried drug loaded and unloaded disks, respectively (3).

#### 2.4.3. Fourier-Transform Infrared (FTIR) Analysis

To prepare samples for FTIR analysis, a pestle and mortar was used to crush the material. The crushed sample was then mixed with Merck IR-spectroscopy-grade potassium bromide in a 1:100 ratio. The resulting mixture was dried at 40 °C and compressed into semitransparent disks with a diameter of 12 mm using a pressure of 60 kN applied for 2 min with a pressure gauge from Shimadzu (Kyoto, Japan). FTIR spectra were recorded using an FTIR spectrometer (FTIR 8400S, Shimadzu) over the wavelength range of 4000–500 cm^−1^ [21].

#### 2.4.4. Scanning Electron Microscopy (SEM)

SEM was performed by using a scanning electron microscope (Hitachi, S 3000 H, Hitachi, Japan) to observe the surface morphology and porosity of quince hydrogels. Samples were mounted on the aluminum mount and coated with gold palladium by using a sputter coater. An accelerating voltage of 10 KV with a working distance of 10–25 mm was applied for scanning [22].

#### 2.4.5. In Vitro Drug Release Measurement

In vitro drug release studies according to the specifications of the United States Pharmacopeia by using a USP apparatus II were performed on hydrogel disks loaded with metoprolol tartrate. A total of 900 mL of relevant dissolution media, i.e., 0.2 M HCl (pH 1.2) and a phosphate buffer of pH 7.4 were used. The media were stirred at 50 rpm at 37 °C. A total of 5 mL of aliquot was removed at 0, 0.5, 1, 2, 4, 6, 8, 10, 14, 18, and 24 h after filtering. A total of 5 mL of fresh medium was added to conserve the volume at each interval. Samples were diluted with the relevant buffer and analyzed at 222 nm using a UV-spectrophotometer (Shimadzu, Japan). The cumulative drug release study was calculated in triplicate and reported as the mean. An univariate ANOVA analysis was performed to evaluate drug release, and the *p*-value was determined [23].

#### 2.4.6. Percentage Drug Release

Percent drug release was determined using following equation:Percentage drug release = *F*_1_*/F_load_ ×* 100(4)
where *F*_1_ shows the amount of metoprolol tartrate released at time t, and *F_load_* represents the quantity of metoprolol loaded in the hydrogel.

#### 2.4.7. Evaluation of Release Kinetics

For the analysis of drug release zero-order, first-order, Higuchi and Korsmeyer–Peppas models are used. The release profile was analyzed by using the semi-empirical power equation by Peppas to gain an insight into the solute release mechanism [24].

#### 2.4.8. In Vivo Evaluation

##### Study Design for In Vivo Evaluation

The study strictly adhered to established scientific protocols for the ethical treatment of laboratory animals, as approved by the Institutional Ethical Committee of the Arid Agriculture University Rawalpindi, Punjab (Approval No. PMAS-AAUR/IEC/525). White albino rabbits, with an average weight of 2.5 ± 0.61 kg, were selected for the study and acclimated to a controlled laboratory environment maintained at a temperature of 25 °C. A total of 24 rabbits were used, divided into two groups of twelve. Each group was individually housed and provided unrestricted access to both water and food. Before the commencement of the study, the rabbits underwent a fasting period of 12 h.

In first group, a commercially available metoprolol tablet was administered at a dose of 12.5 mg, while the second group received a hydrogel equivalent to 12.5 mg of metoprolol. During the fasting period and throughout the experiment, the rabbits had unrestricted access to water. Both formulations, the crushed metoprolol tablet and the hydrogel, were administered to the rabbits using an oral gavage method. The protocols for oral administration involved crushing the tablet and the hydrogel disk and dispersing them in normal saline. The resulting drug dispersion was administered to the first group, while the hydrogel–saline mixture was then carefully administered to the second group via oral gavage. This approach ensured precise and controlled hydrogel administration.

Blood samples (2 mL each) were collected from the marginal ear vein at specified time points during the study. The total blood volume collected from each rabbit did not exceed the established safe bleed limit, which is typically around 6.5–7.5 mL per kg of body weight. These blood samples were processed by centrifugation at 2500 rpm for 5 min, and the resulting plasma was transferred to separate sample tubes and stored at freezing temperatures until analysis. A blank plasma sample (without the drug dose) was also retained for reference. To determine the concentration of metoprolol in the blood plasma samples, high-performance liquid chromatography (HPLC) was employed following a specific analytical method.

##### Standard Preparation

Stock solutions of metoprolol of 1 mg/mL strength were prepared using methanol. This was further diluted to 20–100 ng/mL dilutions and dispersed in blank plasma. These dilutions were then mixed and vortexed with an equal volume of chilled methanol to precipitate plasma proteins. This procedure was repeated three times to effectively remove all the plasma proteins in the sample. All these samples were run on the HPLC to develop a calibration curve for sample analysis.

##### HPLC Analysis

The amount of metoprolol was determined through the HPLC system LC-20AD Shimadzu with photodiode array detector (SPD-M20A-Shimadzu). LC-Solution software was used to analyze the data. Separation was carried out by a C18 column sized at 4.6 mm × 250 mm. The chromatographic conditions used for metoprolol evaluation in plasma are listed in Table 3.

##### In Vivo Pharmacokinetic Evaluation

At different time intervals of 0, 0.5, 1, 2, 4, 6, 8, and 12 h, blood was collected from the marginal ear vein as a sample. The sample was centrifuged at 3500 rpm for 15 min to collect plasma, which was then stored at −20 °C until analysis. These samples were then mixed with an equal volume of methanol, vortexed, and centrifuged three times to remove plasma proteins. These samples were then oven dried and reconstituted with the mobile phase and injected into the HPLC. The plasma was evaluated for estimation of pharmacokinetic parameters (C_max_, T_max_, AUC and t_1/2_).

## 3. Results and Discussion

### 3.1. Percentage Yield and Chemical Reaction Scheme

The percentage yield of CM was found to be 9.3%. The copolymeric hydrogel formulations of CM-co-AA- and CM-co-MAA were synthesized by the free radical copolymerization method. This method was selected, as it is the most desired method for the synthesis of copolymeric hydrogels because it reportedly produced a highly cross-linked hydrogel network with an enhanced degree of cross-linking, better mechanical strength, extraordinary swelling, and sustained release profile. A possible mechanism of the copolymerization of AA onto the CM backbone consists of three steps, i.e., initiation, propagation, and termination presented in Figure 1A,B. 

### 3.2. Swelling Studies

#### 3.2.1. Hydrogel Swelling at Different pHs

The swelling ratio augments with the pH of the surrounding medium as depicted in Figure 2A–C. Drug diffusion and release factors depend upon the swelling response due to the change in pH. A significant increase in swelling ratio was credited by the following factors: Acrylic acid contains a carboxylate ionic group. The monomer pendent group ionization results in swelling. The swelling rate and increasing pH have a direct relation. At higher pHs like 7.4, due to electrostatic repulsion, swelling increases. As the pH increases over the pKa of the polymer and monomer, the complexes become ionized. Negative groups hold repulsive forces at 1.2 pH, which tend to decrease swelling. A similar fact was reported for prepared pH-sensitive thiolated arabinoxylan-grafted acrylic acid copolymer, which supported the fact that hydrogel swelling increases with alkaline pH due to the electrostatic force of repulsion [25].

#### 3.2.2. Effect of Polymer Concentration on Swelling Behavior of CM-co-AA and CM-co-MAA Hydrogels

Effect of polymer concentration on swelling behavior of different formulation was investigated. Swelling ratios were depicted in Figure 2C and Figure 3, respectively. The swelling of hydrogel samples was assessed by the measurement of the liquid amount absorbed by the material as a function of time until saturation. The swelling ratios of *Cydonia oblonga*-mucilage-based hydrogels with increased concentration of polymer were slightly decreased at pH 1.2: for CM-co-AA, it decreased from 4.73 to 4.63; for CM-co-MAA hydrogels, it decreased from 3.28 to 2.99. At pH 7.4, swelling ratios were increased for CM-co-AA from 25.47 to 30.07; and for CM-co-MAA, it increased from 14.01 to 16.29. Swelling of hydrogels was low at acidic pH values and high at basic pH values, indicating the pH-responsive nature of polymer. At acidic pH values, ionic groups were not ionized, repulsive forces were not generated, chains were in a collapsed state, and swelling was low. At basic pH values, repulsive forces were created owing to ionization of ionic groups that resulted in enhanced swelling. Water uptake of hydrogel was influenced by variation in the polymer concentration. *Cydonia oblonga* mucilage is an anionic hydrophilic polymer. As the content of polymer was increased, the hydrogel was highly hydrated owing to ionization of ionizable groups and generation of repulsive forces. The increase in ionic polymers within the hydrogel caused an enhancement in its swelling capacity at basic pH values due to increased chain relaxation as well as osmotic swelling pressure. Porosity of the hydrogel was also increased at high concentrations of the polymer, which is attributable to decreased crosslinking density that resulted in improved swelling. At acidic pH values, there was a slight decrease in the swelling of *Cydonia oblonga*-mucilage-based AA and MAA hydrogels by increasing the polymer concentration, because ionic groups were present in a collapsed state [26].

#### 3.2.3. Effect of Monomer Concentration on Swelling Behavior of CM-co-AA Hydrogels

The swelling behavior of CM-co-AA hydrogels with varying monomer concentrations is depicted in Figure 4. It was observed that with an increase in pH, the swelling ratio of formulations increased drastically. The swelling ratio for M-1 and M-2 at pH 1.2 was 4.72 and 4.25; and at pH 7.4, it was 19.25 and 31.74, respectively. Acrylic acid is an anionic monomer. Carboxylic acid groups in the structure of the hydrogel is responsible for pH-sensitive swelling behavior. Acrylic acid has a pKa value of 4.28; thus, at a pH below 4, chains presented a collapsed state; hence, the swelling ratio was reduced. However, at a pH above 7, the degree of ionization increased due to formation of carboxylate ions present in acrylic acid; repulsion between the networks was caused by these ions, which resulted in a rapid increase in the swelling ratio. At acidic pH by increasing monomer concentration (KPS) swelling ratio was slightly decreased which was caused by presence of more protonated carboxylic groups [27].

#### 3.2.4. Effect of Monomer Concentration on Swelling Behavior of CM-co-MAA Hydrogels

The swelling behavior of CM-co-MAA hydrogels with different methacrylic acid contents was studied. The swelling ratios of hydrogels are given in Figure 5. The swelling ratio of M-3 and M-4 at pH 1.2 was 3.28 and 2.58; and at pH 7.4, it was 10.99 and 16.28, respectively. The degree of swelling of MAA-containing hydrogels was lower at acidic pH values and higher at basic pH values. The reason for this difference was that when the pH of outer environment is higher than the pKa of MAA, carboxylic acid groups are ionized, resulting in more a hydrophilic polymer network that causes rapid absorption of water and a higher swelling ratio. Increased ionization of functional groups causes electrostatic repulsion between the ionized groups, leading to chain expansion, which in turn leads to chain relaxation. Thus, at basic pH values, hydrogels were swelled by a relaxation-controlled mechanism [28].

#### 3.2.5. Effect of Crosslinker Concentration on Swelling Behavior of CM-co-AA and CM-co-MAA Hydrogels

Hydrogel formulations were prepared with varying crosslinker contents to examine the effect of crosslinker concentration on swelling behavior of different formulations. The swelling ratios of CM-co-AA and CM-co-MAA hydrogels with different crosslinker concentrations are depicted in Figure 6 and Figure 7, respectively. Dynamic equilibrium swelling was found to be decreased by increasing the molar content of MBA. At acidic pH values, the swelling ratio for CM-co-AA decreased from 3.77 to 3.71; and for CM-co-MAA, it was decreased from 2.91 to 2.87. At basic pH values, the swelling ratio of hydrogels was reduced for CM-co-AA from 22.28 to 20.66; and for CM-co-MAA, it was reduced from 13.56 to 10.11. The swelling behavior of hydrogels was strongly dependent on the extent of crosslinking. By increasing the crosslinker content’s crosslinking density and the polymeric structure, stability was increased due to a greater number of crosslink points in the polymeric cage, while the porosity of hydrogel was decreased. At lower crosslinking densities, the network was a loose packing with a higher hydrodynamic free volume, so the chains held more solvent molecules, which resulted in higher swelling. This clearly indicated that the crosslinker content has a significant effect on swelling behavior [29].

#### 3.2.6. Drug Loading

The quantity of drugs loaded in disks of all formulations is shown in Figure 8. The amount of drugs loaded in disks was according to swelling of hydrogel disks. The polymer (CM) and monomer (AA) are hydrophilic in nature, which assists the absorption of the drug solution by hybrid formulation. As the polymer content increases from 0.5 to 1 g, drug loading increased (161–170 mg/0.5 gm disk). Similarly, with an increment in AA content in the formulation, the C1–C2 drug loading increased from 140 to 175 mg/0.5 gm disk. Moreover, the drug loading capacity decreased (128 to 115 mg/0.5 gm disk) with an increase in MBA concentration in C1 and C2, respectively. Swelling of CM-co-AA hydrogels was higher than that of CM-co-MAA. So, the amount of drugs loaded was higher in CM-co-AA hydrogels, while formulating a pH-sensitive graphene oxide/poly (N-methylolacrylamide-methyl acrylate) composite hydrogel [30].

#### 3.2.7. FTIR Characterization of Hydrogels

FTIR spectra of (A) MT, (B) CM, (C) AA, (D) MAA, (E) CM-co-AA, (F) CM-co-AA (L), (G) CM-co-MAA, and (H) CM-co-MAA (L) are given in Figure 9. The absorption peak of *Cydonia oblonga* mucilage at 3402 cm^−1^ was attributed to N–H stretch; at 2933 cm^−1^, it was owed to the C–H stretch; 1606 cm^−1^ was for N–H bending; 1409 cm^−1^ was for C–H bending; 1354 cm^−1^ was due to the C–H bending vibration of –CH_2_; and 1041 cm^−1^ was for CH stretching [31]. Absorption peaks of acrylic acid at 3504 cm^−1^ corresponded to O–H stretching; 1639 cm^−1^ was due to C=O stretching; C=O bending in –COOH was at 1452 cm^−1^; and 1319 cm^−1^ was for C–C stretching (11). Characteristic peaks for drug metoprolol tartrate in infrared spectra were present at 3929 cm^−1^ due to O–H; 1595 cm^−1^ because of C=O; and 1242 cm^−1^ for C–N groups. Absorption peaks at 2862 cm^−1^ were due to the C–H group, and those at 1109 cm^−1^ was due to the alkyl aryl ether linkage [32]. Comparing the IR spectrum of *Cydonia oblonga* mucilage with the IR spectrum of *Cydonia oblonga* mucilage-co-acrylic acid, it was observed that the absorption band for *Cydonia oblonga* mucilage at 1409 cm^−1^ was shifted to 1433 cm^−1^, suggesting the grafting reaction of AA on CM. The characteristic stretching vibration peak of hydrophilic groups of *Cydonia oblonga* mucilage at 3400 cm^−1^ was shifted to a lower frequency. This lowering in frequency of hydrophilic groups indicated the presence of hydrogen bonding in the hydrogels. These indications showed that hydrophilic groups of *Cydonia oblonga* mucilage have reacted with –COO^−^ groups of acrylic acid [20]. Characteristic peaks for metoprolol tartrate were also present in the IR spectrum of CM-co-AA-loaded hydrogel at 3911 cm^−1^, 1589 cm^−1^, and 1255 cm^−1^ due to O–H, C=O and C–N groups, respectively. The peak at 2924.00 cm^−1^ was due to the C–H group. No interaction between the drug and polymer was found [6].

The FTIR spectrum of MAA indicated a peak at 2941 cm^−1^ that was due to asymmetric stretching of methyl C–H group. The peak at 1649 cm^−1^ was assigned to a carboxylic acid group, and the peak at 1589 cm^−1^ showed the vibrations of C=C stretching [33]. The IR spectrum of the pure drug metoprolol tartrate displayed characteristic peaks at 3929.00 cm^−1^, 1595.33 cm^−1^, and 1242.09 cm^−1^ due to O–H, C=O and C-N groups, respectively. The peaks of 2862.00 cm^−1^ and 1109.00 cm^−1^ were due to the C–H group and alkyl aryl ether linkage, respectively. In the FTIR spectrum of the CM-co-MAA, there is the appearance of a new characteristic absorption band at 1726 cm^−1^ due to carbonyl stretching vibrations of carboxylic acid groups and at 1448 cm^−1^ due to asymmetric stretching styles of carboxylate anions, respectively, along with other bands with weak intensities, authenticating the formation of the graft copolymer. Further, the formation of CM-co-MAA is supported by a weak intensity band at 3313 cm^−1^ due to stretching vibrations of hydrophilic groups as compared to a FTIR spectrum of the CM (3402 cm^−1^). The weakening of the band is due to the utilization of some hydrophilic groups of CM during the formation of the graft copolymer [34]. Characteristic peaks of drug were also present in the spectrum of the CM-co-MAA (loaded) formulation. Peaks at 1602 cm^−1^ and 1242 cm^−1^ show the presence of C=O and C–N groups. Peaks at 2868 cm^−1^ and 1111 cm^−1^ were due to the C–H group and alkyl aryl ether linkage, respectively [35].

#### 3.2.8. Scanning Electron Microscopy (SEM) of CM-co-AA and CM-co-MAA

SEM analysis was conducted on drug-loaded freeze-dried hydrogel formulations to gain deeper insights into their porous structure and surface morphology (Figure 10). In the case of CM-co-AA freeze-dried hydrogel with metoprolol tartrate, SEM microphotographs (Figure 10A–C) clearly depicted the presence of pores within the hydrogel matrix. These pores play a crucial role in enhancing the hydrogel’s water absorption and retention capabilities. Similarly, SEM micrographs of CM-co-MAA freeze-dried hydrogel loaded with the drug (Figure 10D–F) exhibited a highly porous surface. The presence of this porous structure in drug-loaded hydrogels is a critical aspect to consider. It is expected to have a significant impact on both the rate and extent of hydrogel swelling and, consequently, on drug release kinetics, as has been discussed in prior research [35].

#### 3.2.9. In Vitro Drug Release from CM-co-AA and CM-co-MAA Hydrogels

##### Effect of pH

The percentage drug release for CM-co-AA hydrogels was reduced from 40.21 to 39.49, and for CM-co-MAA, it decreased from 16.41 to 14.41; at a basic pH, the drug release was increased from 75.05 to 80.99 for CM-co-AA hydrogels and from 65.45 to 69.11 for CM-co-MAA hydrogels. Among various graft copolymer formulations prepared with varying contents of polymer, monomer, and crosslinker, M2 presents superior properties with regards to swelling, controlled, and pH-responsive drug release. Drug release was higher at basic pH values as compared to acidic pH values. Drug release was increased at high pH values due to the ionization of ionic groups and the development of repulsive forces between polymer chains and availability of more free volume for penetrant water molecules. At acidic pH values, ionic groups were in non-ionized and collapsed states, and release was lower due to the presence of more originally coiled molecules. At basic pH values, the drug release was enhanced by an increase in the concentration of the polymer. An ANOVA depicted that there is a significant difference in release after 24 h in acidic (1.2 pH) and basic (7.4 pH) conditions, with *p* = 0.0079 [36].

#### 3.2.10. In Vitro Drug Release from CM-co-AA and CM-co-MAA Hydrogels with Different Polymer Concentration

CM-co-AA and CM-co-MAA hydrogels observed an increase in release as presented in Figure 11 and Figure 12, which was attributable to the presence of more hydrophilic groups. By increasing the polymer content, the hydrogel becomes highly hydrated due to the availability of more hydroxyl groups, which, in turn, leads to enhanced drug release [36].

#### 3.2.11. In Vitro Drug Release from CM-co-AA, and CM-co-MAA Hydrogels with Varying Crosslinker (MBA) Concentration

The effect of crosslinker concentration on drug release from CM-co-AA and CM-co-MAA hydrogels is shown in Figure 13 and Figure 14, respectively. The percentage drug release was decreased for CM-co-AA hydrogels from 38.81; and for CM-co-MAA hydrogels, it decreased from 17.85 to 15.92; while at a high pH value, it decreased from 68.37 to 63.21 for CM-co-AA hydrogels; and it decreased from 65.21 to 61.19 for CM-co-MAA hydrogels. Increasing MBA concentration in hydrogels had a negative effect on drug release at low and high pHs. MBA is hydrophobic in nature: by increasing the molar content of MBA, swelling and drug release from the hydrogel decreased. The higher the concentration of crosslinker, the higher the crosslinking density and the tighter the hydrogel structure. Due to generation of more crosslink points, more hydroxyl groups were consumed in crosslinking reactions. As a result, the network spaces are diminished, and less water enters the hydrogel. Increased MBA concentration was beneficial for mechanical stability; however, at the same time, it led to lower porosity [36].

#### 3.2.12. In Vitro Drug Release from CM-co-AA Hydrogels with Varying Concentration of Acrylic Acid (AA)

In vitro drug release data from CM-co-AA hydrogels with varying concentrations of acrylic acid are depicted in Figure 15. At low pH values, drug release from M-1 and M-2 was 40.11 and 39.51; and at high pH values, it was 65.21 and 81.99, respectively. Swelling of the hydrogel in response to pH of the medium played a key role in drug release. The drug release from the hydrogel was also dependent upon the composition of the hydrogel. The action of repulsive forces as well as the ionization of the ionizable functional groups are responsible for enhanced swelling of hydrogel in an alkaline medium. As a result of increased hydration due to electrostatic repulsive forces among charged groups of acrylic acid swelling, drug release was higher. At acidic pHs, electrostatic forces between uncharged carboxyl group vanished and hydration was decreased; thus, swelling reduced, resulting in restricted release of metoprolol tartrate. More ions (COO^−^ group) were present within the network as the amount of acrylic acid was increased, leading to positive osmotic pressure and an increased swelling rate. Generation of electrostatic repulsion due to similar charges of the network chain led to the extensity of network and less Brownian movement, causing molecular relaxation. Increase in the concentration of AA resulted in an increased percent drug release from hydrogel formulation [36].

#### 3.2.13. In Vitro Drug Release from CM-co-MAA Hydrogels with Varying Concentration of MAA

Results of in vitro drug release from CM-co-MAA containing varying monomer concentration are shown in Figure 16. At pH 1.2, drug release from M-3 and M-4 was 17.65, 15.85; and at pH 7.4, it was 62.67 and 69.65, respectively. At pH 7.4, hydrophilicity was increased owing to the deprotonation of ionic groups and widening of originally coiled molecules that were caused by electrostatic repulsion along the chain; at low pH values, carboxylic groups of MAA show less swelling due to the presence of protonated carboxylic groups. At high pH values, expansion of molecules depends on the percent ionization of the carboxylic group. Drug release was enhanced by increasing the pH of the medium. At higher pH (7.4) values, due to more swelling, osmotic pressure inside the gel also increases as compared to lower pH (1.2) values, leading to increased drug release. At high pH values with increasing concentration of MAA, more carboxylic groups of MAA were available for ionization; as a result, network chains moved apart, causing enhanced swelling and more drug release [36].

#### 3.2.14. Drug Release Kinetics

The pattern of drug release was analyzed in buffer solutions of pH 1.2 and pH 7.4. The data obtained at pH 7.4 were placed in zero-order, first-order, Korsmeyer–Peppas, Higuchi, and Hixson–Crowell models for the evaluation of drug release pattern. The most suitable mechanism was elucidated on the base of the best fitness of the release model. The release model could be anticipated by considering the regression value near to 1. The values of regression coefficient for all models from drug-loaded hydrogels with different contents of monomers, crosslinkers, and polymers are depicted in Table 4. Based on the highest regression coefficient value (R^2^), the best fit model for all formulations was found to be the Korsmeyer–Peppas model. Values of the release exponent (n) are also given in Table 3. The value of n for CM-co-AA was greater than 0.45. According to the n value, the drug release mechanism from the CM-co-AA and CM-co-MAA hydrogels was non-Fickian [37].

#### 3.2.15. In Vivo Pharmacokinetic Evaluation

In vitro studies depicted that CM-co-AA showed better a study profile than CM-co-MAA hydrogels. Based on preliminary investigations, these CM-co-AA formulations with maximum in vitro cumulative drug release were selected for in vivo evaluation. The plasma profile of metoprolol tartrate after administration of the hydrogel formulation and the marketed product is shown in Figure 17; this figure clearly identifies a sustained release pattern of metoprolol from the hydrogel network system. Statistically, the plasma concentration vs. time profile of hydrogel formulation was found to be significantly different (*p* < 0.05) than the marketed product in a Student-t paired test. The hydrogel preparation demonstrated a relative bioavailability of 230% compared to the marketed product. Moreover, the pharmacokinetic parameters of both the formulations are listed in Table 5. The value of C_max_ for the drug-loaded formulation was found to be 156.48 ng/mL, and it is 1.63 times higher as compared to C_max_ of marketed metoprolol, which was 96 ng/mL. The value of T_max_ for hydrogels (4 h) was significantly increased compared to that of the marketed product: it manifests a slower rate of drug absorption from hydrogels because of the sustained release pattern of the polymer matrix [38].

MRT of the hydrogel-loaded formulation (8.80 h) was also found to be greater than that of marketed metoprolol (4.91 h). A longer mean residence time delays depletion of the administered dose by maintaining a specific required drug concentration in the body. Moreover, relative bioavailability was calculated by comparing values of the AUC for hydrogel formulation (1322 ng/mL × h) and marketed formulation (574 ng/mL × h): the results depicted that there was 2.3 times greater bioavailability in the hydrogels than the marketed metoprolol formulation. The use of hydrogels not only lessened dose frequency but also ameliorated patient acceptability and a reduced risk of side effects due to the stable absorption pattern. Furthermore, Cydonia Oblanga mucilage indicated an efficient excipient for controlling the release of loaded drug. 

## 4. Conclusions

In conclusion, our study successfully demonstrated the potential of quince (*Cydonia oblonga*)-mucilage-based hydrogels as smart materials for the controlled drug delivery of metoprolol tartarate. The hypotheses that these hydrogels could provide pH-sensitive swelling and controlled drug release were substantiated by results. In vitro drug release studies revealed that both CM-co-AA and CM-co-MAA hydrogels exhibited promising performance in delivering metoprolol tartarate, with sustained release profiles extending up to 24 h. However, it is worth noting that CM-co-AA hydrogels displayed a superior study profile compared to CM-co-MAA hydrogels in terms of drug release characteristics. Furthermore, the evaluation of in vivo parameters further strengthened our findings, indicating the efficacy of these hydrogels as an efficient controlled delivery system for metoprolol tartarate. Hence, our research supports the hypothesis that quince-mucilage-based hydrogels, specifically CM-co-AA, can serve as pH-sensitive, controlled drug delivery systems for metoprolol tartarate, with both in vitro and in vivo results corroborating their potential in pharmaceutical applications.

## Figures and Tables

**Figure 1 pharmaceutics-15-02445-f001:**
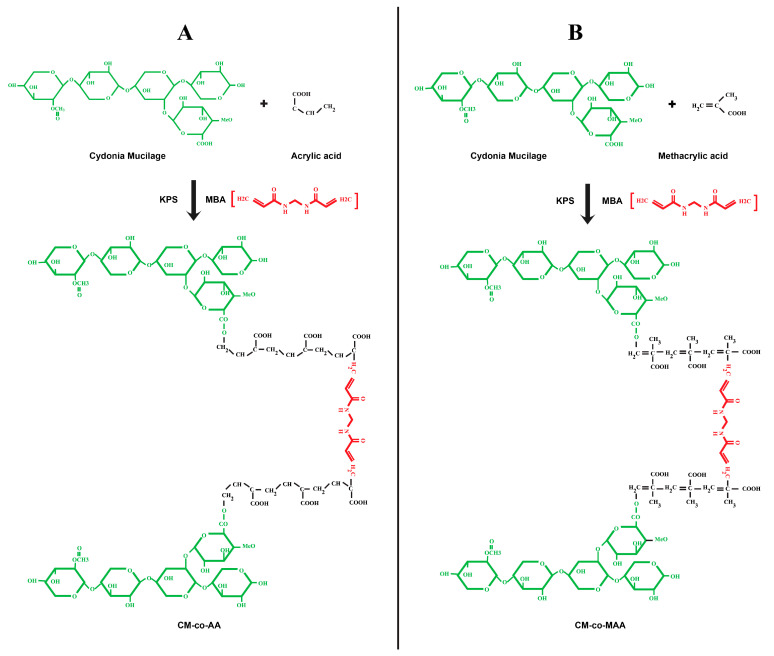
Schematic representation of the development process of the pH-sensitive hydrogel (**A**) Hydrogel swelling at acidic pH (quince seed mucilage-co-AA), (**B**) Swelling of hydrogel at Neutral and alkaline pH (quince seed mucilage-co-MAA).

**Figure 2 pharmaceutics-15-02445-f002:**
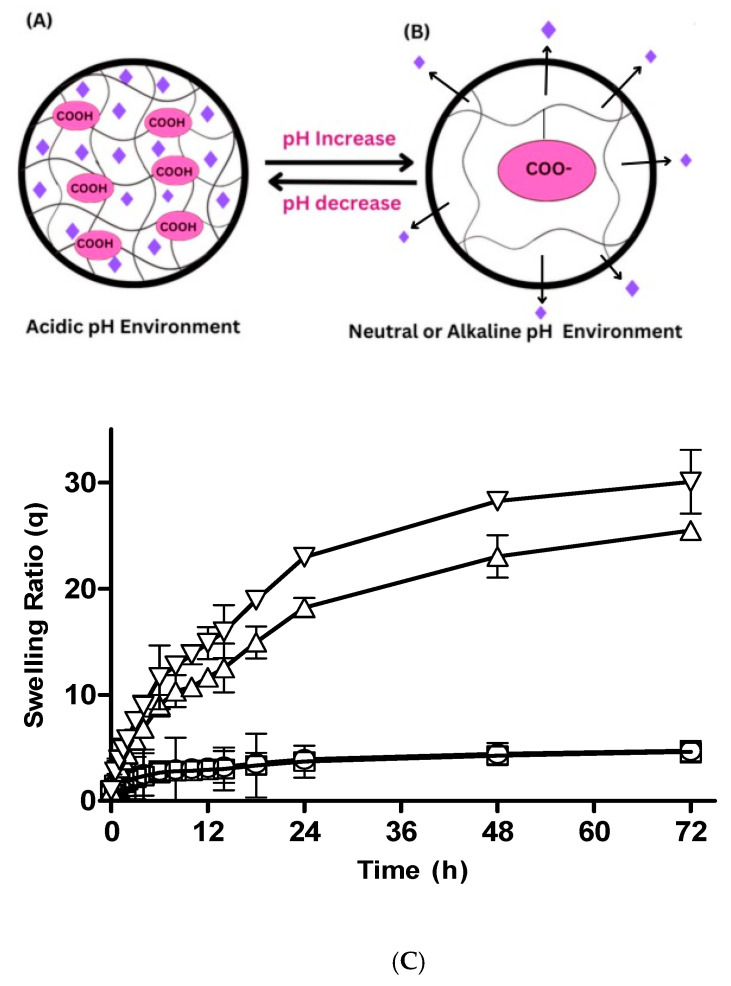
Proposed Mechanism of Hydrogel Swelling at Different pHs. (**A**) is the demonstration of hydrogel swelling at acidic pH, where (**B**) is description of Swelling of hydrogel at Neutral and alkaline pH. (**C**) The swelling ratios of CM-co-AA hydrogels with varying polymer concentration at different physiological pHs. Formulation P-1 at pH 1.2 (◯), P-2 at pH 1.2 (□), P-1 at pH 7.4 (△) and P-2 at pH 7.4 (▽). Indicated values are means (*n* = 3) ± SD.

**Figure 3 pharmaceutics-15-02445-f003:**
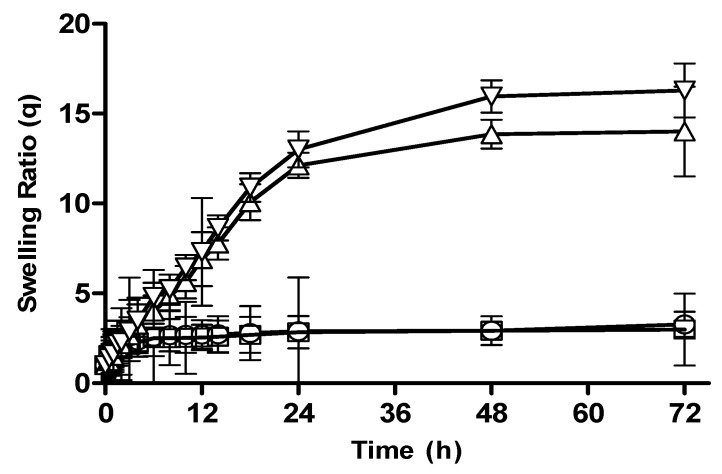
The swelling ratios of CM-co-MAA hydrogels with varying polymer concentration at different physiological pHs. Formulation P-3 at pH 1.2 (◯), P-4 at pH 1.2 (□), P-3 at pH 7.4 (△) and P-4 at pH 7.4 (▽). Indicated values are means (*n* = 3) ± SD.

**Figure 4 pharmaceutics-15-02445-f004:**
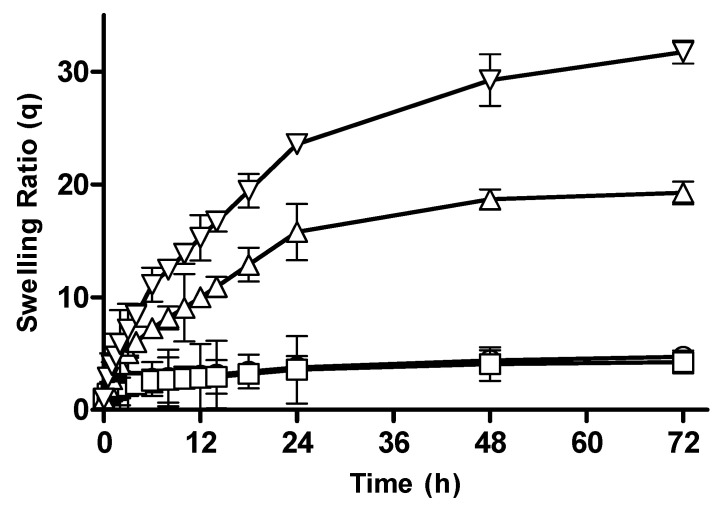
Swelling ratios of CM-co-AA hydrogels with varying monomer concentration at different physiological pHs. Formulation M-1 at pH 1.2 (◯), M-2 at pH 1.2 (□), M-1 at pH 7.4 (△) and M-2 at pH 7.4 (▽). Indicated values are means (*n* = 3) ± SD.

**Figure 5 pharmaceutics-15-02445-f005:**
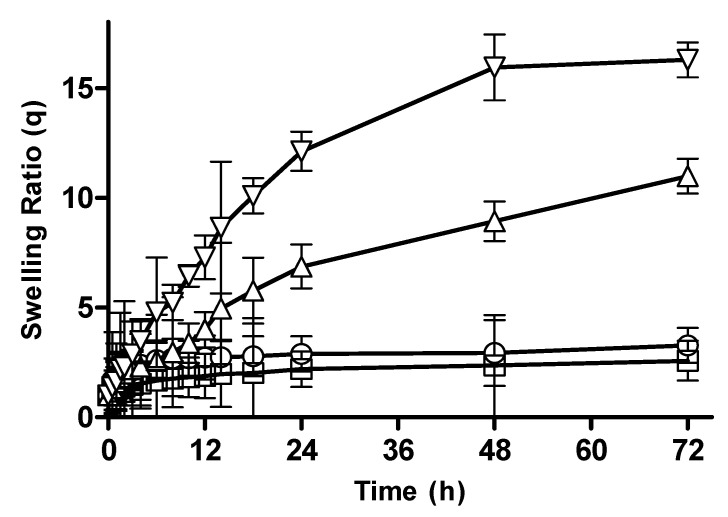
Swelling ratio of CM-co-MAA hydrogels with varying monomer concentration at different physiological pHs. Formulation M-3 at pH 1.2 (◯), M-4 at pH 1.2 (□), M-3 at pH 7.4 (△) and M-4 at pH 7.4 (▽). Indicated values are means (*n* = 3) ± SD.

**Figure 6 pharmaceutics-15-02445-f006:**
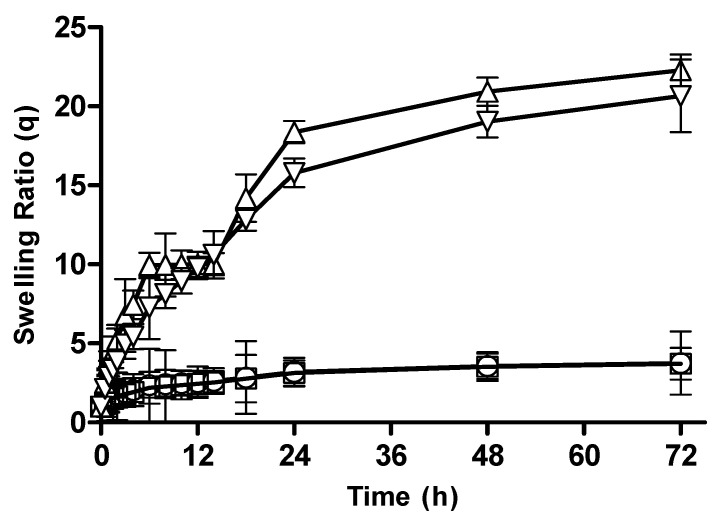
Swelling ratios of CM-co-AA hydrogels with varying concentration of crosslinker at different physiological pHs. Formulation C-1 at pH 1.2 (◯), C-2 at pH 1.2 (□), C-1 at pH 7.4 (△) and C-2 at pH 7.4 (▽). Indicated values are means (*n* = 3) ± SD.

**Figure 7 pharmaceutics-15-02445-f007:**
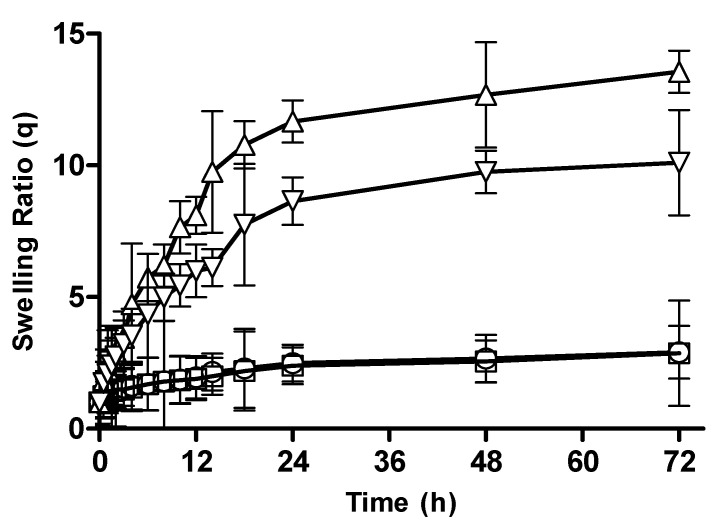
Swelling ratios of CM-co-MAA hydrogels with varying concentration of crosslinker at different physiological pHs. Formulation C-3 at pH 1.2 (○), C-4 at pH 1.2 (□), C-3 at pH 7.4 (△) and C-4 at pH 7.4 (▽). Indicated values are means (*n* = 3) ± SD.

**Figure 8 pharmaceutics-15-02445-f008:**
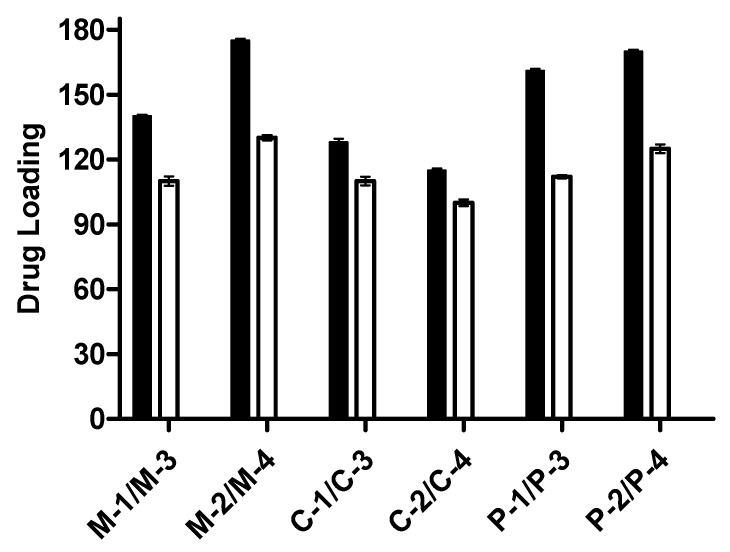
Drug loading into the CM-co-AA hydrogels (black bars) and CM-co-MAA hydrogels (white bars). Indicated values on *y*-axis are mg of drug per gram of disk.

**Figure 9 pharmaceutics-15-02445-f009:**
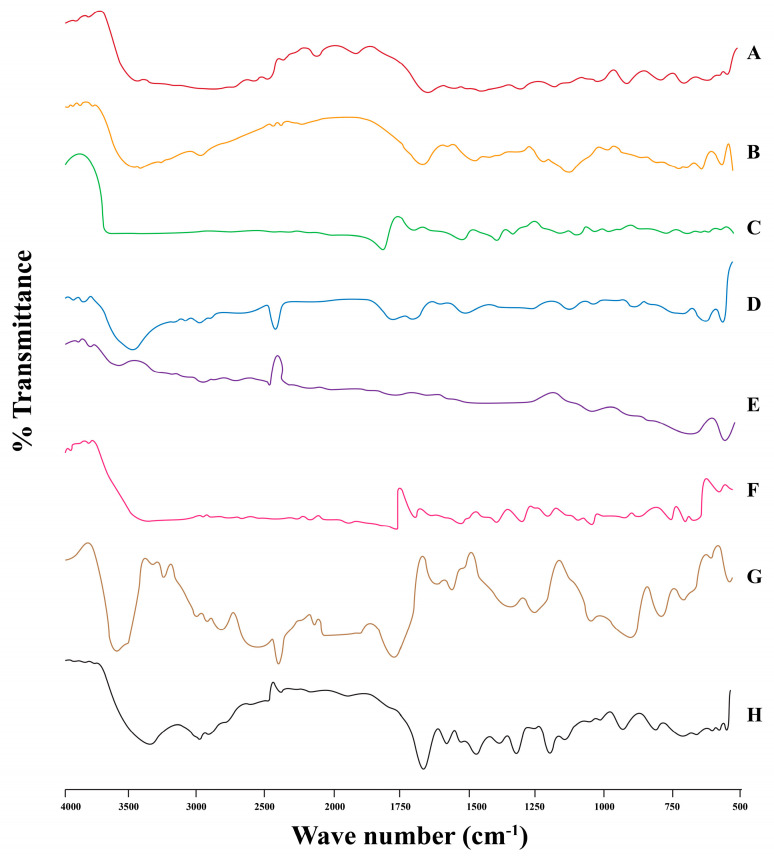
FTIR of (**A**) (metoprolol tartrate), (**B**) (*Cydonia oblonga* mucilage), (**C**) (Acrylic Acid), (**D**) (unloaded CM-co-AA hydrogel), (**E**) (Loaded CM-co-AA hydrogel), (**F**) (Methacrylic Acid), (**G**) (unloaded CM-co-MAA hydrogel), and (**H**) (loaded CM-co-MAA hydrogel).

**Figure 10 pharmaceutics-15-02445-f010:**
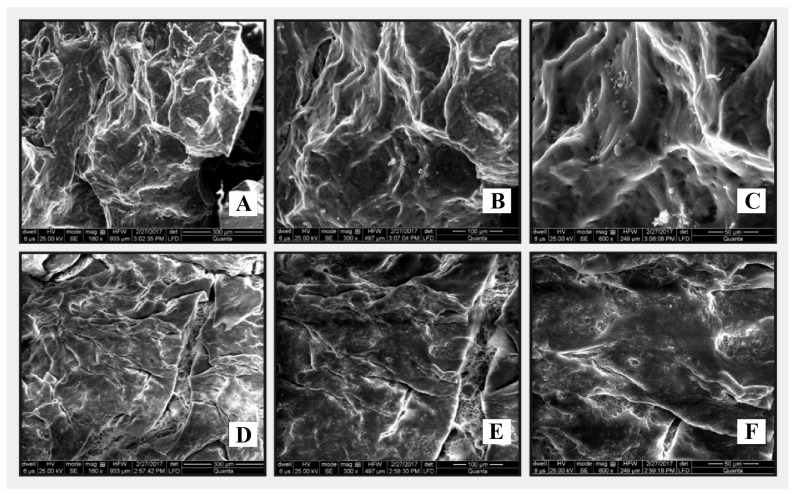
SEM pictographs of CM-co-AA (**A**–**C**) and CM-co-MAA (**D**–**F**) at 160, 300, and 600×.

**Figure 11 pharmaceutics-15-02445-f011:**
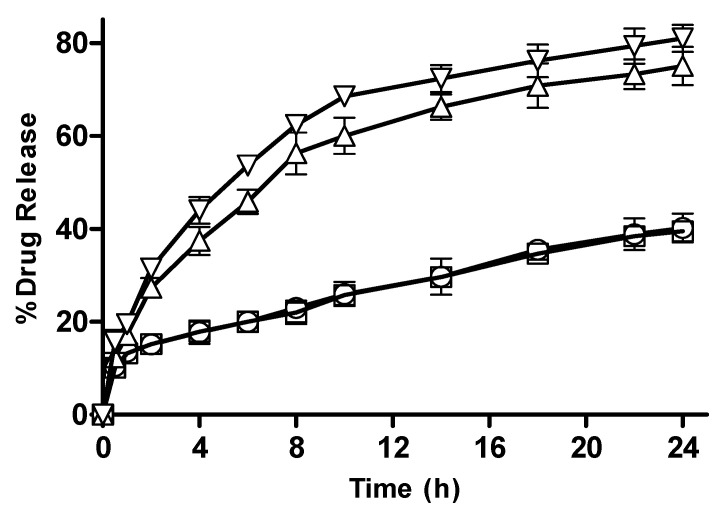
Percentage drug release with different polymer concentration from CM-co-AA hydrogels at different physiological pHs. Formulation P-1 at pH 1.2 (◯), P-2 at pH 1.2 (□), P-1 at pH 7.4 (△) and P-2 at pH 7.4 (▽). Indicated values are means (n = 3) ± SD.

**Figure 12 pharmaceutics-15-02445-f012:**
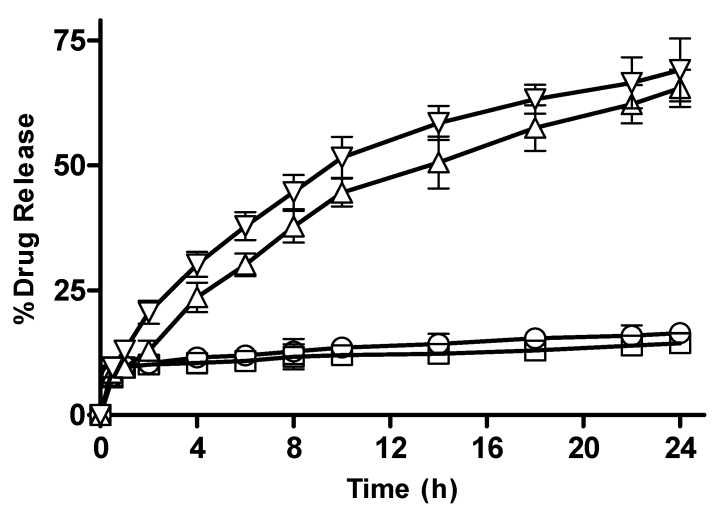
Percentage drug release with different polymer concentration from CM-co-MAA hydrogels at different physiological pHs. Formulation P-3 at pH 1.2 (◯), P-4 at pH 1.2 (□), P-3 at pH 7.4 (△) and P-4 at pH 7.4 (▽). Indicated values are means (n = 3) ± SD.

**Figure 13 pharmaceutics-15-02445-f013:**
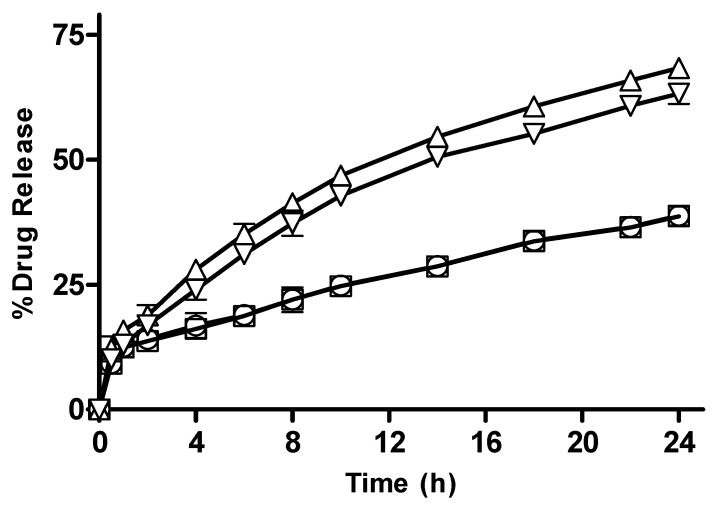
Percentage drug release with varying crosslinker concentration CM-co-AA hydrogels at different physiological pHs. Formulation C-1 at pH 1.2 (◯), C-2 at pH 1.2 (□), C-1 at pH 7.4 (△) and C-2 at pH 7.4 (▽). Indicated values are means (n = 3) ± SD.

**Figure 14 pharmaceutics-15-02445-f014:**
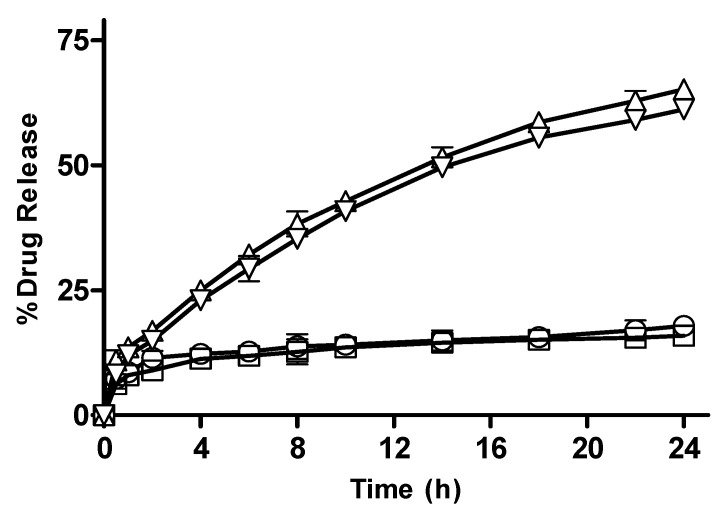
Percentage drug release with varying crosslinker concentration from CM-co-MAA hydrogels at different physiological pHs. Formulation C-3 at pH 1.2 (◯), C-4 at pH 1.2 (□), C-3 at pH 7.4 (△) and C-4 at pH 7.4 (▽). Indicated values are means (n = 3) ± SD.

**Figure 15 pharmaceutics-15-02445-f015:**
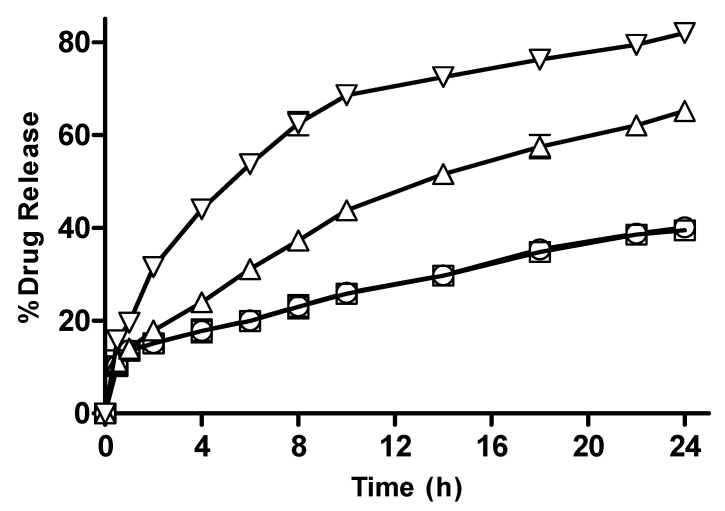
Percentage drug release with varying concentrations of acrylic acid from CM-co-AA hydrogels at different physiological pHs. Formulation M-1 at pH 1.2 (◯), M-2 at pH 1.2 (□), M-1 at pH 7.4 (△) and M-2 at pH 7.4 (▽). Indicated values are means (n = 3) ± SD.

**Figure 16 pharmaceutics-15-02445-f016:**
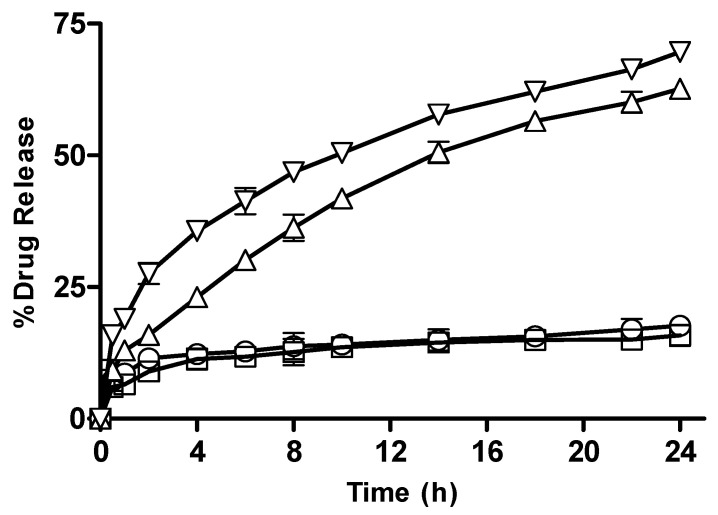
Percentage drug release with varying concentration of acrylic acid from CM-co-MAA hydrogels at different physiological pHs. Formulation M-3 at pH 1.2 (◯), M-4 at pH 1.2 (□), M-3 at pH 7.4 (△) and M-4 at pH 7.4 (▽). Indicated values are means (n = 3) ± SD.

**Figure 17 pharmaceutics-15-02445-f017:**
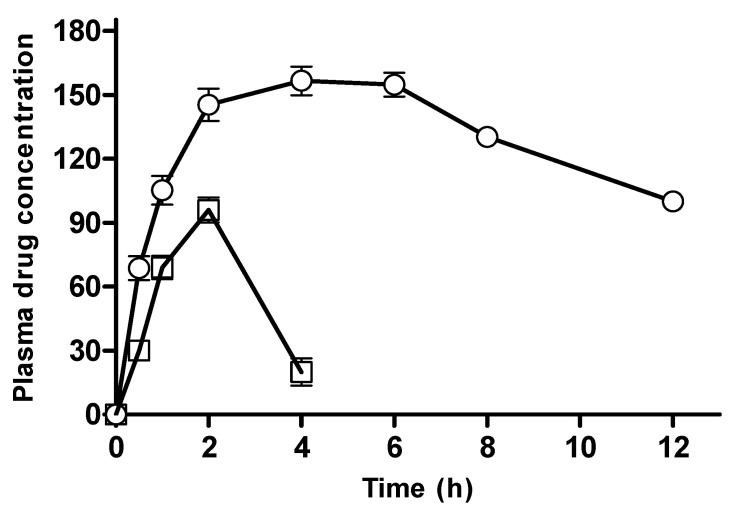
Plasma profile of Metoprolol tartrate after administration of hydrogel formulation (◯) and marketed products (□).

**Table 1 pharmaceutics-15-02445-t001:** Composition of 100 g of *Cydonia oblonga* mucilage-co-Acrylic acid (CM-co-AA) hydrogels.

Formulation Code	Polymer *Cydonia oblonga* Mucilage (g)	Crosslinker MBA (% Mole Ratio of Monomer)	Initiator KPS (% Mole Ratio of Monomer)	Acrylic Acid (g)
M-1	1	0.02	0.02	12.5
M-2	1	0.02	0.02	17.5
C-1	1	0.03	0.02	15
C-2	1	0.04	0.02	15
P-1	0.5	0.02	0.02	15
P-2	1.5	0.02	0.02	15

**Table 2 pharmaceutics-15-02445-t002:** Composition of 100 g of *Cydonia oblonga* mucilage-co-Methacrylic acid (CM-co-MAA) hydrogels.

Formulation Code	Polymer *Cydonia oblonga* Mucilage (g)	Crosslinker MBA (% Mole Ratio of Monomer)	Initiator KPS (% Mole Ratio of Monomer)	Methacrylic Acid(g)
M-3	1	0.02	0.02	30
M-4	1	0.02	0.02	35
C-3	1	0.03	0.02	35
C-4	1	0.04	0.02	35
P-3	0.5	0.02	0.02	35
P-4	1.5	0.02	0.02	35

**Table 3 pharmaceutics-15-02445-t003:** Chromatographic conditions.

**Chromatography**	LC-20AD Shimadzu
**Column**	C 18.5 μ, (250 mm × 4.6 mm)
**Mobile phase**	Acetonitrile:methanol:20 mM ammonium acetate buffer (25:55:20)
**Flow rate**	One milliliter per minute
**Temperature**	Ambient
**Wavelength for detection**	274 nm
**Dilution solvent**	HPLC graded water
**Retention time**	5.53 min

**Table 4 pharmaceutics-15-02445-t004:** Application of drug release models on dissolution data.

Formulation	Zero-Order Model		First-Order Model		Higuchi Model		Korsmeyer–Peppas Model			Hixson–Crowell Model	
code	R^2^	K_0_	R^2^	K_1_	R^2^	kH	R^2^	kKP	n	R^2^	kHC
M1	0.801	3.196	0.94	0.051	0.994	13.321	0.997	12.463	0.525	0.908	0.015
M2	0.613	4.351	0.946	0.105	0.939	18.49	0.943	10.107	0.451	0.892	0.028
M5	0.846	3.092	0.963	0.049	0.984	12.812	0.991	10.616	0.571	0.936	0.014
M6	0.74	3.528	0.926	0.063	0.951	14.79	0.951	14.168	0.516	0.883	0.018
C1	0.79	3.394	0.949	0.258	0.992	14.171	0.993	13.38	0.522	0.913	0.016
C2	0.82	3.24	0.954	0.053	0.99	13.473	0.993	11.997	0.544	0.924	0.015
C5	0.844	3.214	0.966	0.052	0.984	13.321	0.99	11.077	0.569	0.939	0.015
C6	0.861	3.036	0.968	0.047	0.977	12.546	0.987	9.857	0.59	0.943	0.014
P1	0.604	0.333	0.943	0.104	0.937	18.427	0.942	21.177	0.447	0.887	0.028
P2	0.654	3.964	0.933	0.082	0.958	16.784	0.961	18.531	0.463	0.876	0.022
P5	0.771	3.517	0.948	0.062	0.97	14.709	0.971	13.743	0.526	0.909	0.017
P6	0.854	3.186	0.968	0.051	0.98	13.183	0.989	10.649	0.58	0.942	0.015

**Table 5 pharmaceutics-15-02445-t005:** Various Pharmacokinetic parameters.

Parameter	Unit	Hydrogel Formulation	Marketed Product
Lambda_z	1/h	0.094999903	0.0089321
T_max_	h	4	2
C_max_	ng/mL	156.4879899	96
AUC 0-t	ng/mL × h	1321.630847	574.456
MRT 0-inf_obs	h	8.807584284	4.91

## Data Availability

Data are contained in this article.

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
