# Peer review of "Cydonia oblonga-Seed-Mucilage-Based pH-Sensitive Graft Copolymer for Controlled Drug Delivery—In Vitro and In Vivo Evaluation"

_pharmaceutics, 2023, doi:10.3390/pharmaceutics15102445_

Round 1

Reviewer 1 Report

The manuscript presents results of the investigation of the potential of Quince seed mucilage as an excipient in a graft copolymer for an oral controlled drug delivery system.

The work contains several interesting results, but I think should be majorly revised to improve its quality and fit the journal scope.

1) The Introduction should be update to show more clearly the perspective and advantages of using Quince seed mucilage for preparation of novel polymeric hydrogels. What is the reason for that, why not to take any other source for natural biopolymers for their cross-linking? What is the benefit of this current approach? The same motivation should be also included in discussion section and conclusion.

Also, information on other drug delivery systems and investigation of their in vivo evaluations should be added as it is mentioned even in manuscript title. Following references can be used, for example:

- Zhu YH , Wang JL , Zhang HB , Khan MI , Du XJ , Wang J . Incorporation of a rhodamine B conjugated polymer for nanoparticle trafficking both in vitro and in vivo. Biomater Sci. 2019;7(5):1933-1939. doi: 10.1039/c9bm00032a.

- Kuskov A, Nikitovic D, Berdiaki A, Shtilman M, Tsatsakis A. Amphiphilic Poly-N-vinylpyrrolidone Nanoparticles as Carriers for Nonsteroidal, Anti-Inflammatory Drugs: Pharmacokinetic, Anti-Inflammatory, and Ulcerogenic Activity Study. Pharmaceutics. 2022;14(5):925. doi: 10.3390/pharmaceutics14050925.

- Yellepeddi VK, Joseph A, Nance E. Pharmacokinetics of nanotechnology-based formulations in pediatric populations. Adv Drug Deliv Rev. 2019;151-152:44-55. doi: 10.1016/j.addr.2019.08.008.

2) Cydonia oblonga mucilage preparation, purification and characterization section should be improved (chromatography, IR and NMR spectroscopy) in order to show that extracted and used base for copolymers can be identified qualitatively and quantitatively properly. The characterization of biopolymers must prove technological approach declared.

3) Scanning Electron Microscopy (SEM) results discussion should be improved. SEM microphotographs of freeze-dried hydrogel without drug in this case show the presence of pores in the synthesized hydrogel samples, but these pores are not the pores of hydrogel. These pores are defects and cavities as residues of ice crystals, eliminated in the process of freeze-drying. For such systems SEM of dried samples can not be prove of their porosity.

4) Results of In-Vivo pharmacokinetic evaluation should be improved. The proper statistic evaluation methods should be applied. P – values must be estimated, to show significance of obtained pharmacokinetic parameters. The example of pharmacokinetic studies results presentation can be found in references mentioned above.

Minor editing of English language required through the text.

Author Response

Dear Reviewer 1: The point by point response to your valuable comments are given below. The manuscript with track changes is attached here with as an attachment.

Reviewer’s Evaluation

Response and Revisions

The Introduction should be update to show more clearly the perspective and advantages of using Quince seed mucilage for preparation of novel polymeric hydrogels. What is the reason for that, why not to take any other source for natural biopolymers for their cross-linking? What is the benefit of this current approach? The same motivation should be also included in discussion section and conclusion.

Also, information on other drug delivery systems and investigation of their in vivo evaluations should be added as it is mentioned even in manuscript title. Following references can be used, for example:

- Zhu YH , Wang JL , Zhang HB , Khan MI , Du XJ , Wang J . Incorporation of a rhodamine B conjugated polymer for nanoparticle trafficking both in vitro and in vivo. Biomater Sci. 2019;7(5):1933-1939. doi: 10.1039/c9bm00032a.

- Kuskov A, Nikitovic D, Berdiaki A, Shtilman M, Tsatsakis A. Amphiphilic Poly-N-vinylpyrrolidone Nanoparticles as Carriers for Nonsteroidal, Anti-Inflammatory Drugs: Pharmacokinetic, Anti-Inflammatory, and Ulcerogenic Activity Study. Pharmaceutics. 2022;14(5):925. doi: 10.3390/pharmaceutics14050925.

- Yellepeddi VK, Joseph A, Nance E. Pharmacokinetics of nanotechnology-based formulations in pediatric populations. Adv Drug Deliv Rev. 2019;151-152:44-55. doi: 10.1016/j.addr.2019.08.008.

Cydonia oblonga seed mucilage majorly composed of glucuronic acid and xylose which is a biocompatible polysaccharide. Cydonia oblonga seed mucilage is water-swellable so potentially acts as smart material.  It usually de-swells in acidic conditions and swells in water or basic buffers. In present work these novel characteristics augmented by crosslinking with acrylic acid and methacrylic acid. 

Papers have been cited as advised.

.

Cydonia oblonga mucilage preparation, purification and characterization section should be improved (chromatography, IR and NMR spectroscopy) in order to show that extracted and used base for copolymers can be identified qualitatively and quantitatively properly. The characterization of biopolymers must prove technological approach declared.

Purification of Cydonia oblonga mucilage purification and characterization:

Upon addition of Molish’s reagent in extract followed by addition of sulphuric acid, the ring was observed at the junction, which confirmed the carbohydrates presence Carbohydrate were marked as present, amino acids and tannins were absent while fats were removed by n-Hexane during extraction process. Hence purity of Cydonia oblonga mucilage extract was confirmed. FTIR spectrum indicated absorption peak of Cydonia oblonga mucilage at 3402 cm1attributed to N-H stretch, at 2933 cm−1was owing to C-H stretch, 1606 cm−1 was for N-H bending, 1409 cm−1 was for C-H bending, 1354 cm−1 was due to C-H bending vibration of –CH2, 1041 cm−1 was for CH stretching

Scanning Electron Microscopy (SEM) results discussion should be improved. SEM microphotographs of freeze-dried hydrogel without drug in this case show the presence of pores in the synthesized hydrogel samples, but these pores are not the pores of hydrogel. These pores are defects and cavities as residues of ice crystals, eliminated in the process of freeze-drying. For such systems SEM of dried samples can not be prove of their porosity.

The SEM analysis is conducted on the loaded hydrogel, not unloaded hydrogel. In the manuscript, there was an error, which has since been identified and corrected. Moreover, regarding the pores in hydrogels,

we greatly value the reviewer's comments. However, we'd like to offer a possible explanation for the presence of pores in our SEM images. These pores are not mere anomalies or defects but are intricately linked to the hydrogel's fundamental structure. This assertion finds support in various scientific principles and observations.

Firstly, during the synthesis of hydrogels, a important aspect of the process involves the formation of a three-dimensional network of polymer chains through cross-linking. This process inherently results in the creation of voids or pores within the hydrogel matrix. The size and distribution of these pores are closely tied to factors such as the choice of polymer, cross-linking density, and reaction conditions. Consequently, it is entirely expected for hydrogel samples to exhibit porosity.

Secondly, it's important to note that the hydrogel in our study goes beyond mere structural integrity; it acts as a drug delivery system. The successful incorporation of drug molecules into the hydrogel matrix hinges on the existence of these pores. The pores serve as reservoirs in which drug molecules are housed, facilitating controlled drug release. This mechanism not only relies on but is also greatly influenced by the porous nature of the hydrogel.

Lastly, the significance of these pores extends to the heart of our research - the controlled release of drugs. These pores function as channels through which drug molecules are gradually released, allowing for precise control over drug release kinetics. Without these pores, achieving the desired level of controlled drug release within the hydrogel matrix would pose both scientific and practical challenges.

In conclusion, we respect the reviewer's comments, and we hope this explanation sheds light on the presence of pores in our SEM images.

Results of In-Vivo pharmacokinetic evaluation should be improved. The proper statistic evaluation methods should be applied. P – values must be estimated, to show significance of obtained pharmacokinetic parameters. The example of pharmacokinetic studies results presentation can be found in references mentioned above

The authors are thankful for raising this concern that gives the opportunity to improve the in-vivo results presentation. The missing standard error of means (SEM) have been added to fig. 17.  The plasma concentration Vs time profile of hydrogel formulation was compared with the marketed product in student-t paired test and it was found significantly different (p < 0.05). This lead us to calculating the relative bioavailability for the hydrogel formulations and found to be 230% compared to the marketed product. All the findings have been added in the newer version of the manuscript.

Reviewer 2 Report

The paper CYDONIA OBLONGA SEED MUCILAGE BASED pH SENSITIVE GRAFT COPOLYMER FOR CONTROLLED DRUG DELIVERY; IN VITRO AND IN VIVO EVALUATION by Sarfraz et al can bwe accepted for the publication in Pharmaceutics after a major revision concerning the following points:

1)      The most part of the acronyms througout all the document from the abstract to the conclusions are not explained;

2)      The state of the art with specific regard to the treated issue is not exaustively presented, due to the fact that Quince Seed Mucilage has already presented, starting at least from 2018, as stimuli-responsive smart biopolymer and these papers are not cited;

3)      A schematic representation of the development process of the pH-sensitive hydrogel should be given. The swelling response due to change in pH should be graphically represented too;

4)      A representetion of the structure formula of metoprolol could allow a comprehension of the mechanism of pH-sensitive release

5)      The following paper: Mater. Chem. Front., 2023, 7, 216–229 should be cited due to the factthat it presents a pH-sensitive drug delivery mechanism.

Author Response

Dear Reviewer 2: The point by point response to your valuable comments are given below. The manuscript with track changes is attached here with as an attachment.

Reviewer:2

Reviewer’s Evaluation

Response and Revisions

 The most part of the acronyms throughout all the document from the abstract to the conclusions are not explained;

All acronyms have been explained.

The state of the art with specific regard to the treated issue is not exhaustively presented, since Quince Seed Mucilage has already presented, starting at least from 2018, as stimuli-responsive smart biopolymer and these papers are not cited;

Papers have been cited as advised.

A schematic representation of the development process of the pH-sensitive hydrogel should be given. The swelling response due to change in pH should be graphically represented too;

The schematic representation of the development process has been given in paper. Proposed mechanism of hydrogel swelling due to change in pH has been represented.

 A representation of the structure formula of metoprolol could allow a comprehension of the mechanism of pH-sensitive release

The combination of natural polymers with synthetic monomer resulted in development of a non-toxic and mechanical strong crosslinked network of hydrogel, which prolonged the release of drug. Our research work evaluated pH-sensitive hydrogels based on Cydonia oblonga mucilage with acrylic acid and methacrylic acid. Acrylic acid and methacrylic acid are synthetic monomers, used broadly in both biomedical and pharmaceutical fields. These are pH-sensitive by nature because of their functional groups (COOH), which enhance their sensitivity to the external environment. Due to its high swelling capability in basic medium, these are employed highly in the development of pH-responsive hydrogels. So basic mechanism of drug release from hydrogels is the pH responsive swelling of hydrogel.

 The following paper: Mater. Chem. Front., 2023, 7, 216–229 should be cited because it presents a pH-sensitive drug delivery mechanism.

Papers have been cited as advised.

Reviewer 3 Report

In this study, Muhammad Sarfraz et al. prepared quince seed mucilage incorporated hydrogels through free radical polymerization with acrylic acid and methacrylic acid as monomers, and N, N-methylene bisacrylamide as crosslinkers. Utilization of the hydrogel as a pH response delivery vehicle for oral delivery of metoprolol tartrate was demonstrated with varying monomer-, crosslinker-, and mucilage-concentrations. The topic is interesting, however, there are several issues that should be addressed to improve the manuscript.

Comments:

  1. The rationale of the hydrogel design should be clarified, why acrylic acid and methacrylic acid were used as monomer? Hydrogels prepared through free radical polymerization of these monomer are not biodegradable. Can the hydrogel be eliminated after oral administration?
  2. There is a lack of evidence that demonstrates the advantages of incorporating quince seed mucilage in the hydrogel, as the performance of hydrogels without quince seed mucilage was not investigate in this study.
  3. Protocols for oral administration of the hydrogel in the in vivo study should be presented with more detail. Did rabbits swallow the hydrogel discs by themselves?
  4. Line 29: abbreviations must be defined when mentioned for the first time.
  5. Line 21: what does the dynamic swelling ratio and equilibrium swelling ratio mean?
  6. Line 261: why increasing monomer (KPS) concentration was associated with more protonated carboxylic groups?
  7. The quality of Figure 8 should be improved.
  8. In Figure 9, it is suggested that the authors showed morphologies of drug-loaded hydrogels in addition to the unloaded ones.
  9. The subtitles at lines 356 and 372 are almost the same.

None

Author Response

Dear Reviewer 3: The point by point response to your valuable comments are given below. The manuscript with track changes is attached here with as an attachment. 

Reviewer’s Evaluation

Response and Revisions

The rationale of the hydrogel design should be clarified, why acrylic acid and methacrylic acid were used as monomer? Hydrogels prepared through free radical polymerization of these monomer are not biodegradable. Can the hydrogel be eliminated after oral administration?

The selection of acrylic acid (AA) and methacrylic acid (MAA) as monomers in the hydrogel design is driven by their pH-responsive properties, which are particularly advantageous for drug delivery applications. These monomers contain carboxylic acid groups that can undergo protonation and deprotonation in response to changes in pH. This property allows the hydrogel to exhibit a pH-sensitive behavior, which is essential for controlled drug release within the gastrointestinal tract.

The pH-responsive nature of AA and MAA enables the hydrogel to swell and release the drug in a pH-dependent manner. In the acidic environment of the stomach, where these monomers remain protonated, the hydrogel undergoes reduced swelling and exhibits minimal drug release. However, as the hydrogel progresses to the more alkaline environment of the small intestine, deprotonation occurs, leading to increased swelling and enhanced drug release. This precise pH-controlled drug release is vital for optimizing the bioavailability of drugs and ensuring their therapeutic efficacy.

While it is true that hydrogels prepared from AA and MAA are not inherently biodegradable, their intended use in drug delivery systems is focused on controlled and sustained drug release rather than complete degradation.

After drug release, the hydrogel can undergo changes in its physical properties, becoming less crosslinked and more susceptible to swelling. This facilitates its passage through the gastrointestinal tract without causing obstruction or harm. Therefore, the hydrogel can be naturally eliminated after fulfilling its drug delivery function, making it a safe and effective platform for oral controlled drug delivery applications.

There is a lack of evidence that demonstrates the advantages of incorporating quince seed mucilage in the hydrogel, as the performance of hydrogels without quince seed mucilage was not investigate in this study.

The current study, despite its limitations, offers valuable insights and lays the foundation for further research into the utilization of quince seed mucilage in hydrogel-based drug delivery systems. Several factors support the significance of this study.

Firstly, it can be viewed as exploratory research, pioneering the investigation of quince seed mucilage as an excipient in drug delivery. This novel approach delves into the potential of a specific natural material, opening avenues for its application in pharmaceutical sciences.

The study's focus on quince seed mucilage is particularly noteworthy. Natural excipients like mucilage hold unique properties that can bring benefits to drug delivery systems. By exploring the potential of quince seed mucilage, the study contributes not only to our understanding of this specific excipient but also to the broader field of pharmaceutical sciences.

The research also provides valuable findings regarding the synthesis and characterization of quince seed mucilage-based hydrogels. These insights into the pH-sensitive swelling behavior and drug release patterns of these hydrogels are pertinent for designing controlled drug delivery systems, which is a significant advancement.

Furthermore, the inclusion of in vivo pharmacokinetic assessments represents a practical dimension of the study. This step extends beyond laboratory experiments to evaluate how the hydrogel performs within a living organism. Such in vivo data is essential for translating laboratory findings into real-world applications.

In summary, while acknowledging the benefits of a comparative analysis, the present research serves as a valuable starting point in the exploration of quince seed mucilage's potential in pharmaceutical formulations. Its pioneering nature, focus on a unique excipient, valuable findings, practical assessments, and promising results collectively contribute to our understanding of mucilage-based hydrogels for drug delivery systems.

Protocols for oral administration of the hydrogel in the in vivo study should be presented with more detail. Did rabbits swallow the hydrogel discs by themselves?

Following lines have been added for clarity.

The study strictly adhered to established scientific protocols for the ethical treatment of laboratory animals, as approved by the Institutional Ethical Committee at Arid Agriculture University Rawalpindi, Punjab (Approval No. PMAS-AAUR/IEC/525). White albino rabbits, with an average weight of 2.5 ± 0.61 kg, were selected for the study and acclimated to a controlled laboratory environment maintained at a temperature of 25°C. A total of 24 rabbits were used, divided into two groups of twelve. Each group was individually housed and provided unrestricted access to both water and food. Before the commencement of the study, the rabbits underwent a fasting period of 12 hours.

In first group, a commercially available metoprolol tablet was administered at a dose of 12.5 mg, while the second group received a hydrogel equivalent to 12.5 mg of metoprolol. During the fasting period and throughout the experiment, rabbits had unrestricted access to water.

Both formulations, the crushed metoprolol tablet, and the hydrogel, were administered to the rabbits using an oral gavage method. The protocols for oral administration involved crushing the tablet and hydrogel disc and dispersing in normal saline. The resulting drug dispersion was administered to the first group while hydrogel-saline mixture was then carefully administered to the second group via oral gavage. This approach ensured precise and controlled hydrogel administration.

Blood samples (2 ml each) were collected from the marginal ear vein at specified time points during the study. The total blood volume collected from each rabbit did not exceed the established safe bleed limit, which is typically around 6.5–7.5 ml per kg of body weight. These blood samples were processed by centrifugation at 2,500 rpm for 5 minutes, and the resulting plasma was transferred to separate sample tubes and stored at freezing temperatures until analysis. A blank plasma sample (without the drug dose) was also retained for reference. To determine the concentration of metoprolol in the blood plasma samples, high-performance liquid chromatography (HPLC) was employed following a specific analytical method.

Line 29: abbreviations must be defined when mentioned for the first time.

Correction has been done as advised.

Line 21: what does the dynamic swelling ratio and equilibrium swelling ratio mean?

In the context of polymer hydrogels, two significant terms come into play: the "dynamic swelling ratio" and the "equilibrium swelling ratio." These terms are essential for assessing how hydrogels interact with solvents, typically water.

The "dynamic swelling ratio" pertains to the degree of swelling or expansion of the hydrogel at a particular point in time during the swelling process. It provides a snapshot of how much solvent the hydrogel has absorbed at that specific moment. However, it's important to note that the dynamic swelling ratio does not indicate the hydrogel's maximum swelling capacity; instead, it offers insights into the rate at which the hydrogel is absorbing the solvent, making it a valuable tool for tracking changes in swelling over time.

Conversely, the "equilibrium swelling ratio" represents the ultimate level of swelling a hydrogel can achieve under defined conditions and over an extended period. This ratio reflects the point at which the hydrogel has absorbed the maximum amount of solvent possible and has reached a stable equilibrium state with its surroundings. At this equilibrium, the rate of solvent absorption is balanced by the rate of solvent loss, resulting in a constant, unchanging swelling ratio.

These two measures, the dynamic swelling ratio and the equilibrium swelling ratio, are indispensable for understanding the swelling behavior and capabilities of hydrogels in various applications, ranging from drug delivery systems to tissue engineering and biomaterial science. They provide critical insights into how hydrogels interact with their environment and their potential utility in specific contexts.

Line 261: why increasing monomer (KPS) concentration was associated with more protonated carboxylic groups?

The association between increasing initiator (potassium persulfate, KPS) concentration and more protonated carboxylic groups observed in the study can be explained by the role of the initiator in the polymerization process and the resulting increase in polymer chain formation.

In free radical polymerization, the initiator (KPS in this case) plays a crucial role in initiating the polymerization reaction. Initiators generate free radicals, which are highly reactive species with unpaired electrons. These radicals are essential for initiating the polymerization of monomer units (such as acrylic acid and methacrylic acid) and propagating the formation of polymer chains.

When the initiator concentration is increased, there are more molecules of KPS available to generate free radicals. This abundance of free radicals enhances the initiation and propagation of polymerization reactions. As a result, more monomer units undergo polymerization to form polymer chains. During this polymerization process, the carboxylic groups (-COOH) on the monomer units can become protonated, meaning they gain an additional hydrogen ion (H+), transitioning from their acidic form (-COOH) to the conjugate base form (-COO-).

Therefore, the increase in initiator (KPS) concentration leads to a higher generation of free radicals, which, in turn, promotes greater polymer chain formation, ultimately resulting in more protonated carboxylic groups in the polymer structure.

The quality of Figure 8 should be improved.

Quality of Figure 8 has been improved

In Figure 9, it is suggested that the authors showed morphologies of drug-loaded hydrogels in addition to the unloaded ones.

In Figure 9, it appears that there has been an inadvertent mistake in both the figure itself and its accompanying description, as it displays the morphologies of drug-loaded hydrogels, contrary to what was stated as unloaded.

Correction has ben done as follow:

SEM analysis was conducted on drug-loaded freeze-dried hydrogel formulations to gain deeper insights into their porous structure and surface morphology (Fig 9). In the case of CM-co-AA freeze-dried hydrogel with Metoprolol tartarate, SEM microphotographs (Fig 9a, b &c) clearly depicted the presence of pores within the hydrogel matrix. These pores play a crucial role in enhancing the hydrogel's water absorption and retention capabilities. Similarly, SEM micrographs of CM-co-MAA freeze-dried hydrogel loaded with the drug (Fig 9e, f &g) exhibited a highly porous surface. The presence of this porous structure in drug-loaded hydrogels is a critical aspect to consider. It is expected to have a significant impact on both the rate and extent of hydrogel swelling and, consequently, on drug release kinetics, as has been discussed in prior research [26].

The subtitles at lines 356 and 372 are almost the same.

Subtitles are revised to avoid confusion.

Reviewer 4 Report

Dear Author,

As reviewer i have few comments 

1.Why author selected BCS-1 drug?

2.Metoprolol tartrate (MT) is a cardio-selective competitive beta-1 adrenergic receptor antagonist so how this polymeric system beneficial? This approach increases uptake of gel into cell, but our target is on cell surface not inside cell? How author think this approach?

3.In the introduction author should include other marked formulation of metoprolol?

4.Routes of administration for PK study?

5.Conclusion should be answer to hypothesis followed by supporting results?

6.Introduction needs to rewrite?

Author Response

Dear Reviewer 4: The point by point response to your valuable comments are given below. The manuscript with track changes is attached here with as an attachment.

Reviewer’s Evaluation

Response and Revisions

Why author selected BCS-1 drug?

The rationale behind the choice of a BCS class 1 drug, Metoprolol tartrate (MT), lies in its favorable pharmaceutical properties. Being classified as a BCS class 1 drug implies that MT possesses both high solubility and permeability characteristics. This classification is of paramount significance in the realm of drug formulation and delivery. High solubility ensures that a sufficient amount of the drug can readily dissolve in the gastrointestinal fluids, facilitating absorption, while high permeability implies that MT can efficiently traverse biological membranes, further enhancing its bioavailability. Consequently, MT serves as an excellent model drug for the development of controlled-release formulations, as its pharmacokinetic behavior is predictable and reproducible.

Moreover, MT's pharmacological profile aligns with the therapeutic goals of controlled release. As an antihypertensive agent and a cardio-selective competitive beta-1 adrenergic receptor antagonist, MT exerts its effects by acting on the cardiovascular system. However, its relatively short half-life and limited bioavailability due to first-pass metabolism necessitate the design of controlled-release formulations. These formulations can prolong drug release, maintaining therapeutic levels in the bloodstream for an extended duration, and ultimately enhancing the drug's efficacy in managing cardiovascular conditions. Therefore, the selection of MT as the model drug for this study is underpinned by its pharmaceutical attributes and its alignment with the objectives of controlled drug delivery research in the context of cardiovascular treatments.

Metoprolol tartrate (MT) is a cardio-selective competitive beta-1 adrenergic receptor antagonist so how this polymeric system beneficial? This approach increases uptake of gel into cell, but our target is on cell surface not inside cell. How author think this approach?

The polymeric system described in the study offers significant advantages for the controlled delivery of Metoprolol tartrate (MT), a cardio-selective beta-1 adrenergic receptor antagonist. Using quince seed mucilage-based graft copolymer, the system achieves controlled drug release, ensuring a gradual and sustained release of MT. This controlled release is essential in maintaining the drug's therapeutic efficacy while minimizing potential side effects associated with sudden fluctuations in drug levels. Additionally, the pH-sensitive behavior of the system allows it to respond to the varying pH conditions of the gastrointestinal tract, optimizing drug absorption and bioavailability.

The pharmacokinetic evaluation results indicate improved bioavailability, with a higher Cmax and AUC compared to the marketed MT formulation, suggesting that this polymeric system enhances the drug's effectiveness in the bloodstream. Overall, the study demonstrates that quince seed mucilage-based graft copolymer can serve as a smart material for developing an oral controlled release drug delivery system, particularly beneficial for drugs with specific cardiovascular targets like MT.

The polymeric system's primary target is the bloodstream, where Metoprolol tartrate exerts its cardio-selective effects. While the system may increase the uptake of the drug, its primary function is to provide controlled and sustained drug release. This controlled release ensures that therapeutic levels of MT are consistently maintained in the bloodstream, precisely targeting the drug's intended cardiovascular actions.

Routes of administration for PK study

The protocols for the oral administration of the hydrogel in the in vivo study involved crushing the hydrogel discs and dispersing them in normal saline. Subsequently, the hydrogel-saline mixture was administered using an oral gavage technique. This approach ensures precise and controlled administration of the hydrogel and provides clarity regarding the methodology employed in the study.

Conclusion should be answer to hypothesis followed by supporting results

Conclusion has been revised as advised.

In conclusion, our study successfully demonstrated the potential of Quince (Cydonia oblonga) mucilage-based hydrogels as smart materials for controlled drug delivery of metoprolol tartarate. The hypothesis that these hydrogels could provide pH-sensitive swelling and controlled drug release was substantiated by results. In-vitro drug release studies revealed that both CM-co-AA and CM-co-MAA hydrogels exhibited promising performance in delivering metoprolol tartarate, with sustained release profiles extending up to 24 hours. However, it is worth noting that CM-co-AA hydrogels displayed a superior study profile compared to CM-co-MAA hydrogels in terms of drug release characteristics. Furthermore, the evaluation of in vivo parameters further strengthened our findings, indicating the efficacy of these hydrogels as an efficient controlled delivery system for metoprolol tartarate. Hence, our research supports the hypothesis that Quince mucilage-based hydrogels, specifically CM-co-AA, can serve as pH-sensitive, controlled drug delivery systems for metoprolol tartarate, with both in-vitro and in-vivo results corroborating their potential in pharmaceutical applications.

Introduction needs to rewrite?

Introduction has been revised as advised.

Round 2

Reviewer 1 Report

Authors adressed all my comments properly. Manuscript can be accepted after minor text proof and correction.

The only more advice is to mark significant difference of pharmacokinetic parameters difference between studied preparation and marketed product by "*" as usual.

Reviewer 2 Report

The revised version of the manuscript can be accepted in the present form.

Reviewer 4 Report

Dear Author,

This paper revised carefully so recommend accepting it.